# Adolescent Toothbrushing and Its Association with Sociodemographic Factors—Time Trends from 1994 to 2018 in Twenty Countries

**DOI:** 10.3390/healthcare11243148

**Published:** 2023-12-12

**Authors:** Apolinaras Zaborskis, Aistė Kavaliauskienė, Sharon Levi, Riki Tesler, Elitsa Dimitrova

**Affiliations:** 1Faculty of Public Health, Medical Academy, Lithuanian University of Health Sciences, LT-44307 Kaunas, Lithuania; 2Faculty of Odontology, Medical Academy, Lithuanian University of Health Sciences, LT-44307 Kaunas, Lithuania; aiste.kavaliauskiene@lsmu.lt; 3Department of Health Systems Management, School of Health Sciences, Ariel University, Ariel 4077625, Israel; sharonle@ariel.ac.il (S.L.); rikite@ariel.ac.il (R.T.); 4Department of Health Systems Management, The Max Stern Yezreel Valley College, Yezreel Valley, Afula 1930600, Israel; 5Institute for Population and Human Studies, Bulgarian Academy of Sciences & Plovdiv University Paisii Hilendarski, 1000 Sofia, Bulgaria; e.dimitrova@iphs.eu

**Keywords:** adolescents, toothbrushing, dental public health, sociodemographic factors, time trends, HBSC

## Abstract

Regular toothbrushing is the primary self-care method to prevent the most common dental diseases and is considered an important public health indicator. This retrospective observational study aimed to examine cross-national time trends in adolescent toothbrushing behaviour and its association with sociodemographic factors between 1994 and 2018. We studied data from 20 countries that conducted seven surveys of the Health Behaviour in School-aged Children (HBSC) study. Adolescents aged 11, 13 and 15 years responded to questions on their toothbrushing frequency, family affluence and structure. Altogether, reports of 691,559 students were analysed using descriptive statistics and binominal logistic regression. The findings showed an overall positive trend in the prevalence of more-than-once-a-day toothbrushing frequency during the entire study period mainly due to a noticeable increase from 1994 to 2010 (except Denmark and Sweden); this trend continued significantly thereafter in 12 of 20 countries. Across all countries, girls and adolescents from more affluent families were more likely to brush their teeth regularly. These relationships remained unchanged throughout the study period, whereas the age-related difference in toothbrushing prevalence decreased noticeably, and the negative relationship between toothbrushing and living in a non-intact family became evident. While the prevalence of regular toothbrushing among adolescents has increased in recent decades, it is still far behind the recommended level of twice-daily brushing for everyone across all countries. The promotion of toothbrushing needs to start at an early age, with a special focus on boys and adolescents from low-affluence and non-intact families.

## 1. Introduction

Improvement of toothbrushing habits as a primary method to promote oral hygiene is essential for preventing the most common dental diseases [1,2,3]. Furthermore, the frequency of toothbrushing is a valid measure of children’s oral hygiene [4] and may function as an important public health indicator, as it has been shown to be positively associated with good health [5]. The promotion of the prevalence of toothbrushing, using an effective toothpaste twice daily, is recognized in many communities worldwide as a central strategy in oral health programs to reduce the burden of dental caries [6].

Toothbrushing is generally adopted during childhood and adolescence. Dental care should begin and become regular when a child’s first tooth erupts [7]. In this respect, the main influence on this healthy behaviour comes from the family, which is affected by several social characteristics, such as family socioeconomic status (SES), parental education/oral health literacy and family environment and structure. A growing body of research has confirmed the effect of these characteristics on child health and behaviour, which may also have implications for toothbrushing [8,9,10,11,12,13]. The studies showed evidence that children from high-SES families are significantly more likely to brush their teeth than individuals from low-SES families [10,13]. The researchers suggested that these differences may be caused not only by barriers to the affordability of oral hygiene aids but also by other systemic reasons [9]. It may be implied that parents from low-SES households are less likely to follow a healthy lifestyle and favourable self-care, making oral hygiene a lower priority [14].

The Health Behaviour in School-aged Children (HBSC) study, a World Health Organization cross-national project, provides a unique opportunity to explore adolescent health behaviour, including toothbrushing habits, in the social context (see the study website [15]). The HBSC dataset allows for comparisons of adolescent health behaviours across almost all European countries and North America (currently, the study is conducted in 51 countries/regions). As the HBSC study has been conducted over a 30-year period, it also allows for an analysis of trends in adolescent lifestyles, during a period in which many important changes have taken place in Europe as well as around the world [16]. Within each HBSC study wave, the prevalence of regular (more-than-once-a-day) toothbrushing has been compared between countries and reported in the study reports [17,18,19,20,21,22,23], as well as in international journals [12,13]. Large differences in the prevalence of regular toothbrushing were found between countries, yet in every country, it was higher among girls than among boys. In some countries, the prevalence of more-than-once-a-day toothbrushing increased with increasing age, whereas in others, it declined. A decade ago, the trend in toothbrushing from 1994 to 2010 was investigated by Honkala et al. [24] using the HBSC data from 20 countries. Trends have also been examined in individual countries [25,26]. The studies confirmed that the prevalence of the recommended toothbrushing frequency increased in most of the countries, and the differences between countries diminished during the period of observation. These findings raise the question of how the toothbrushing status of adolescents in these countries has changed over the past decade. Moreover, we explore whether the above-described toothbrushing association with demographic and social factors changed during the entire HBSC study period.

The aim of this study was to examine trends in the prevalence of more-than-once-a-day toothbrushing among adolescents in countries/regions that completed all seven consecutive HBSC surveys between 1994 and 2018. The specific objective was to assess the changes in the association of toothbrushing with demographic (gender and age) and social (family affluence and structure) factors. We hypothesized that the trends in recommended toothbrushing prevalence among adolescents in selected HBSC countries would increase over the study period while maintaining the same relationship between toothbrushing and sociodemographic factors.

## 2. Materials and Methods

### 2.1. Study Population

A retrospective observational study design was applied in the current work. We used data from seven waves of HBSC surveys that were conducted between the 1993/1994 school year (coded as 1994) and the 2017/2018 school year (coded as 2018) spanning almost three decades. In all surveys, data collection followed the standard methodology outlined in the HBSC protocols (see, for example, [27]). The surveys included country-representative samples of 11-, 13- and 15-year-olds. Data collection was conducted using a self-report questionnaire administered in schools. Additional details about the aims, theoretical framework and survey methodology of the HBSC study can be found online [15] and in international reports of each survey [17,18,19,20,21,22,23].

Time-trend analysis was performed using data from 20 countries/regions that conducted the survey in each of the seven HBSC study waves between 1994 and 2018. These countries/regions were Austria, Flemish and French-speaking Belgium, Canada, the Czech Republic, Denmark, Estonia, Finland, France, Germany, Greenland, Hungary, Latvia, Lithuania, Norway, Poland, Russia (European part), Scotland, Sweden and Wales (Figure 1). Hereinafter, the Flemish- and French-speaking regions of Belgium, as well as the Scotland and Wales regions of the United Kingdom, were considered independently as countries. Altogether, reports of 691,559 students who answered the questions on their toothbrushing behaviour and sociodemographic characteristics were included in this analysis: 86,238 students in 1994; 83,705 students in 1998; 97,518 students in 2002; 101,891 students in 2006; 107,617 students in 2010; 103,458 students in 2014; and 111,132 students in 2018 study. The overall response rate to the toothbrushing question across the survey waves was 99.3%. The data were obtained via the HBSC Data Management Centre (Bergen University, Norway). The HBSC protocol includes gaining ethics approval in each study country. As our study is based on secondary analysis of the datasets in which respondents did not participate directly, it was deemed unnecessary to obtain further ethical approval.

### 2.2. Variables

Study variables in this analysis included gender and age group as covariates, the number of the survey wave, family affluence and structure as predictors, and toothbrushing frequency as an outcome variable. Only those subjects who had non-missing data for all these variables were included in the analysis.

The variable “survey wave” was coded as 1 = 1994, …, 7 = 2018. Gender was 1 for boys and 2 for girls. By age, students were grouped into 3 groups: 11 years (aged 10 years and 6 months to 12 years and 5 months), 13 years (aged 12 years and 6 months to 14 years and 5 months) and 15 years (aged 14 years and 6 months to 16 years and 5 months).

Family affluence was assessed using the Family Affluence Scale, which was specially developed and validated for the HBSC study [28]. In order to make it as uniform as possible in all study waves, we used four items: (1) number of cars, (2) having your own bedroom, (3) number of computers and (4) number of travels/holidays abroad (the third and fourth items and the fourth item were not mandatory in the scale for surveys in 1994 and 1998, respectively). A family affluence score (FAS) was calculated by summing the points of the responses to these items. Higher FAS values indicated higher family affluence, thus, in accordance with the HBSC reports [19,20], it was used to classify the respondents into three country-specific groups. The first group included 20% of respondents living in the lowest-affluence families (reference group), the second group included 60% of respondents living in medium-affluence families and the third group included 20% of respondents living in the highest-affluence families.

Family structure (FS) was a dichotomous variable that reflected an intact family (child living in a family with both biological parents) and a non-intact family (child living in any other family). This measure was determined using a set of questions about family structure and residence [27].

The mandatory question about the frequency of toothbrushing was validated in the HBSC survey in 1994 [17] and remained consistent across all surveyed countries and subsequent waves of the study. It was formulated as follows: “How often do you brush your teeth?” with the following possible answers: more than once a day; once a day; at least once a week but not daily; less than once a week; never. For analyses, we dichotomized the responses into two categories: More-than-once-a-day and less often (combined with the last four options).

### 2.3. Data Analysis

We conducted quantitative data analysis utilizing the whole sample (data combined from all 20 countries) as well as its subsamples by country. The prevalence (%) of more-than-once-a-day toothbrushing among 11-, 13- and 15-year-old boys and girls was presented for each country and seven consecutive HBSC surveys. Pearson’s χ^2^ test was used for the evaluation of the statistical significance in prevalence between groups. The regularities found in the whole sample were verified in the corresponding subsamples of each country. Binominal logistic regression (BLR) modelling was performed to examine the strength of the association between sociodemographic characteristics (independent variables) and the outcome variable of more-than-once-a-day toothbrushing. The linearity of the trend in regular toothbrushing prevalence by years of the survey was assessed by including the independent variable “survey wave” in the BLR model. This variable was treated as a continuous variable; hence, its odds ratio (OR) estimates the average interwave increase/decrease in the likelihood of regular toothbrushing for each four-year period. Analyses of the whole sample were conducted using individual records, weighting them by the country sample size in each survey wave in order to equalize the importance of data for each country on summary characteristics.

Data were analysed using SPSS (version 21; IBM SPSS Inc., Chicago, IL, USA, 2012). The cut-off level for statistical significance was set at 0.05. Since the current analysis operated with large samples, confidence intervals (CI) were defined at the 99% level.

## 3. Results

Figure 2 shows the crude proportions of students who reported more-than-once-a-day toothbrushing by survey year and HBSC country that completed all seven consecutive HBSC surveys between 1994 and 2018. During the entire study period, there was a particularly noticeable increase (approximately 20 percentage points or more) in the prevalence of toothbrushing behaviour in Lithuania (from 30.0 to 50.8%), Finland (from 37.9 to 65.1%), Belgium (Flemish) (from 42.7 to 63.7%) and France (from 58.2 to 76.1%). In Sweden, Denmark and Norway, countries where the prevalence of the recommended toothbrushing habit was already high in the first survey wave, there were no discernible changes in the trend for the indicator during the study period. Other country-based patterns in the adolescent toothbrushing trend were unclear. Based on the figure, observed differences between the selected countries decreased consistently during this time period. 

In order to further describe the trends, we estimated their linearity. To compare the linearity in different periods, the entire observation period was split into two time intervals, 1994–2010 and 2010–2018. The results of this analysis are shown in Table 1. Considering the first period (1994–2010), a significant positive linear increase in the prevalence of more-than-once-a-day toothbrushing was observed in most of the selected countries, except Denmark and Sweden, in which the proportion of students who report that they brush their teeth regularly significantly decreased but still remained at a high level, above 75%. In Austria, Estonia, Finland and Lithuania, an average interwave (four-year period from one survey to the next survey) increase in the odds of the recommended toothbrushing habit was over 20%. During the last 8 years (2010–2018), the increase in the prevalence of the recommended toothbrushing habit was still observed in many countries, but at a lower value and significance. During this period, adolescents from 12 of 20 countries demonstrated significant positive progress in their toothbrushing. When evaluating the entire study period (1994–2018), it can be concluded that there was still a significant linear increase in the prevalence of toothbrushing in the countries participating in the HBSC study except Denmark and Sweden. In other countries, such as Lithuania, Finland, Belgium (Flemish) and France, an interwave increase in the odds of the recommended toothbrushing was approximately 20% on average during the entire study period.

Across all selected countries and each survey year between 1994 and 2018, girls report a higher prevalence of more-than-once-a-day toothbrushing frequency than boys (Table 2). Regardless of the study year, the OR for girls compared to boys was around 2.0 (Table 3). For girls in most countries, the proportion of respondents reporting the recommended toothbrushing habit increased significantly with age, and this regularity remained in all study waves, but there was no such pattern for boys. Simultaneously, for both genders, the increasing trend from 1994 to 2018 was stronger among younger than older adolescents (see Table 2). For this reason, the age-dependent gradient of toothbrushing decreased; for example, comparing 15-year-olds with 11-year-olds, OR decreased from 1.26 (99% CI: 1.20; 1.32) in 1994 to 0.95 (0.89; 0.98) in 2018 (see Table 3).

Greater family affluence was found to be significantly associated with more frequent toothbrushing in all countries and survey waves except the Czech Republic, Denmark, Greenland, Norway and Poland in 1994 and Finland in 2002. Apart from the survey conducted in 1994, the odds of more-than-once-a-day toothbrushing were almost twice as high among adolescents with high FAS than among adolescents with low FAS, and this indicator had no clear tendency to change over the years of the study (see Table 3). According to the data from the surveys conducted in 1994, 1998 and 2002, in most countries, no significant relationship between toothbrushing frequency and family structure was found. In subsequent years, the surveys have shown that in a growing number of countries, adolescents living in non-intact families were less likely to brush their teeth regularly than those living in intact families. Thus, pooled analysis of the data from the 2018 survey in all countries (see Table 3) showed OR = 0.85 (0.81; 0.88) indicating that adolescents living in non-intact families were 1.18 times less likely to brush their teeth regularly than those living in intact families.

## 4. Discussion

The current study aimed to examine cross-national time trends of toothbrushing behaviour and its association with sociodemographic factors between 1994 and 2018 among adolescents aged 11-, 13- and 15 years. On the basis of available data from 20 countries, the overall positive trend in the prevalence of adolescent more-than-once-a-day toothbrushing was revealed during the entire study period from 1994 to 2010, mainly due to its noticeable increase from 1994 to 2010 (except in Denmark and Sweden) and continued thereafter in many countries but to a lesser degree. Across all countries, girls and adolescents from more affluent families were more likely to brush their teeth regularly. These relationships remained unchanged throughout the study period, whereas the age-related difference in toothbrushing prevalence decreased and the negative relationship between toothbrushing and living in a non-intact family became evident.

Linear trends found in our study confirmed the study hypothesis that the prevalence of recommended toothbrushing among adolescents in selected HBSC countries had an increasing tendency over the study period. A similar study of the same 20 HBSC countries reported a positive trend between 1994 and 2010 [24]. Our study appeared to be an extension of the previous research, supplementing it with new data on the changes in toothbrushing habits during the last 8 years in 2010–2018 and confirming a continuation of positive trends in the majority of selected HBSC countries. This positive result may reflect the success of national oral health policies and initiatives that have been integrated into health-promoting schools since the 1990s [29]. Oral health education is considered to be one of the fundamentals in oral health promotion [30] to achieve a target of twice-daily toothbrushing for everyone [31].

Gender has been recognized as one of the strongest factors explaining differences in adolescent toothbrushing behaviour [17,18,19,20,21,22,23]. Our findings are in line with this previous research, and we further identified that the difference in appropriate toothbrushing frequency between boys and girls stayed almost constant over the years. While age was found to be a significant factor among girls, no significant effect of age was identified among boys. Moreover, due to the fact that boys and girls of younger age groups showed a faster positive time trend in toothbrushing frequency, the difference between age groups decreased. In our study, these changes were more noticeable than in previous studies where trends were examined over a shorter period [24,25,26].

Consistent with previous research [13,26,32,33], we found that adolescents living in low-affluence families were significantly more likely to report infrequent toothbrushing. Hence, it was confirmed that family affluence is a strong factor associated with toothbrushing frequency throughout the entire study period. This should be a major concern since low perceived family wealth is often associated with low SES, which has been identified as an unfavourable social factor leading to an increased risk of oral diseases and lower oral health-related quality of life [34,35,36,37].

While living with both biological parents may explain inequalities in adolescent general health and life quality [38,39] and may affect several risk behaviours [40,41,42], only a few studies have examined this possible role of family structure in oral health-related behaviour [43]. Previous studies found that the association between family characteristics, such as the absence of one of the parents, and toothbrushing behaviours of children appeared to be rather weak and inconsistent [13]. Our recent study on family structure dynamics from 1994 to 2018 in HBSC countries revealed that the proportion of adolescents living in intact families decreased from 80% to 70% [44]. We hypothesized that such a significant change at that time could influence the relationship between family structure and oral healthcare. HBSC data, which were collected over a long period of time, supported this hypothesis, namely, data from the last decade showed that adolescents living with a single parent or in a reconstructed family were significantly less likely to brush their teeth regularly. The differentiating effect of family structure may contribute to parental involvement in child toothbrushing habits, as single parents may be more constrained and experience more difficulties in assisting children to take care of their oral hygiene [45,46]. The lower frequency of toothbrushing among adolescents from non-intact families may be explained by the lower level of oral health education and parental support carried out by only one parent [33,43]. Indirectly, this result may also be a consequence of lower SES in non-intact families as has been shown by socioeconomic studies [47]. To the best of our knowledge, this is the first study to consider the association between family structure and toothbrushing habits among adolescents. The study also suggests that the observed SES differences in toothbrushing in children could reflect the effect of the education of parents, their oral health literacy and health-related role models in families of dental hygiene and dental care; future research may consider these topics for further in-depth study.

The findings of this study highlight inequalities in oral health behaviour between countries as well as between sociodemographic groups within countries. Evidence suggests that inequalities also exist in the prevalence of oral diseases between and within countries [48,49]. Many publications report policy reflections for reducing inequalities in oral health [48,49]. Policies to promote oral health within the healthcare and education systems may be effective in improving oral health [50]. In Germany, children receive instruction on proper oral hygiene as part of group and individual preventive programs in the community and educational institutions, starting from kindergarten until 18 [51]. In Israel, as of 2010, dental health services for children are included in the public health services under the National Health Insurance Law [50]. These services include preventive and restorative dental treatments as well as health education classes in all public schools [52,53].

Based on the findings of the present study, several policy implications that may be common to participating countries can also be suggested. First of all, the study implies that the promotion of toothbrushing needs to start at an early age, with a special focus on boys and adolescents from low-affluence and non-intact families. Next, HBSC findings suggest that poor toothbrushing habits are often accompanied by other health-detrimental behaviours such as regular smoking, unhealthy eating habits and low levels of physical activity, which are common risk factors for several noncommunicable diseases [32,54]. Consequently, toothbrushing as the main dimension of oral health promotion can be easily integrated into general health promotion, school curricula and activities. Finally, investing more in the health education of children and their parents with a focus on toothbrushing will reduce the prevalence of oral diseases and contribute to the overall health of children and adolescents [55].

### Strengths and Limitations

A key strength of the current study lies in the study methodology, which was standardized across countries conducting HBSC surveys and remained unchanged with respect to the variables used in this study throughout the long period of the HBSC study.

Another no less important strength is analysing data from the large, nationally representative samples of adolescents in 20 countries. These countries represent the continents of Europe and North America. Since the prevalence trends of adolescent toothbrushing in most of these countries have similar characteristics, there is an opportunity to generalize the study findings to the countries of both continents. The uniqueness of this study is determined by the fact that time trends in the relationship between adolescent toothbrushing habits and sociodemographic factors, including family wealth and structure, were described for the first time.

Several limitations of the present study should be considered. First, the data were limited by reliance on self-report, which may have been affected by recall and social desirability bias. However, the HBSC team minimized this methodological bias by ensuring that data collection was confidential and anonymous, and the questions were pre-tested on national and international levels before conducting the actual survey. In addition, the study was limited by repeated cross-sectional surveys, which preclude causality. The observed associations may also be affected by the failure to adjust the analysis for other important variables that were unavailable in our dataset, such as parental education level.

Second, family affluence in the 1994 and 1998 surveys was assessed using only two and three items, respectively, of the four-item Family Affluence Scale that was created for the HBSC study [28]. We attempted to reduce this limitation by transforming the FAS into a country-specific three-level score (low, medium and high) representing the lowest 20%, the medium 60%, and the highest 20% of the FAS values, respectively; however, despite this transformation, the association between toothbrushing and FAS in the 1994 and 1998 surveys should be viewed with caution.

Third, in this paper, we analysed only one component of oral health-related behaviour, namely toothbrushing. The data from the HBSC study allows for the examination of additional oral health-related factors, such as smoking or the consumption of sweets, soft drinks and alcohol. Further research should focus on examining the trends in these associated factors to help inform future interventions. Finally, while data from many countries demonstrated an increase in adolescent toothbrushing frequency, we were not able to relate these changes to oral health promotion interventions that have been implemented in participating countries. It also remained unknown to what extent those changes correlated with improvements in adolescent oral health.

Research to investigate changes in policy and programming that may be related to changes in toothbrushing behaviour during the study period and in the future is recommended. Factors related to these programs, impeding or improving toothbrushing habits of children, as well as associations with unhealthy eating habits, cigarette smoking and alcohol consumption should also be explored. In future cross-national studies, a multilevel data analysis method may be applied, which would help evaluate the impact of both individual sociodemographic factors and the national level of healthcare in each country on the studied trend.

## 5. Conclusions

The findings of our study emphasize that the prevalence of more-than-once-a-day toothbrushing frequency among adolescents has increased in recent decades; however, it is still far behind the recommended level of twice-daily brushing for everyone across different countries. Gender and family affluence level remained the most important factors associated with the likelihood of toothbrushing, whereas the age-related difference in toothbrushing prevalence decreased and the negative relationship between toothbrushing and living in a non-intact family became evident. Therefore, the study implies that the promotion of toothbrushing needs to start at an early age, with a special focus on boys and adolescents from low-affluence and non-intact families. In sum, our research underscores the need for a comprehensive, inclusive and targeted approach to oral health promotion and policy, one that recognizes the nuances of gender, family dynamics and socio-economic factors in shaping toothbrushing habits among adolescents. By embracing these insights, we can take meaningful steps towards ensuring better oral health for all, transcending boundaries and improving overall well-being.

## Figures and Tables

**Figure 1 healthcare-11-03148-f001:**
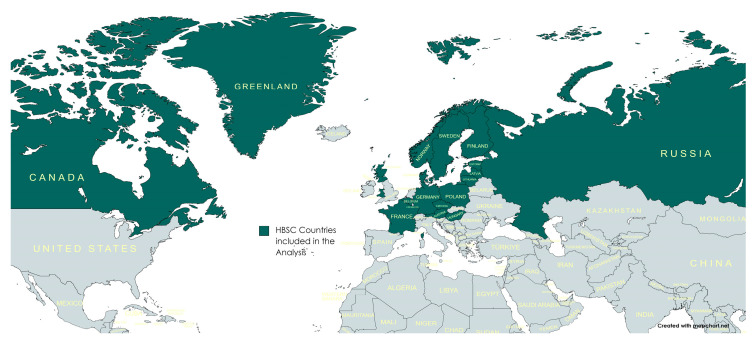
Map of HBSC countries included in the cross-national analysis.

**Figure 2 healthcare-11-03148-f002:**
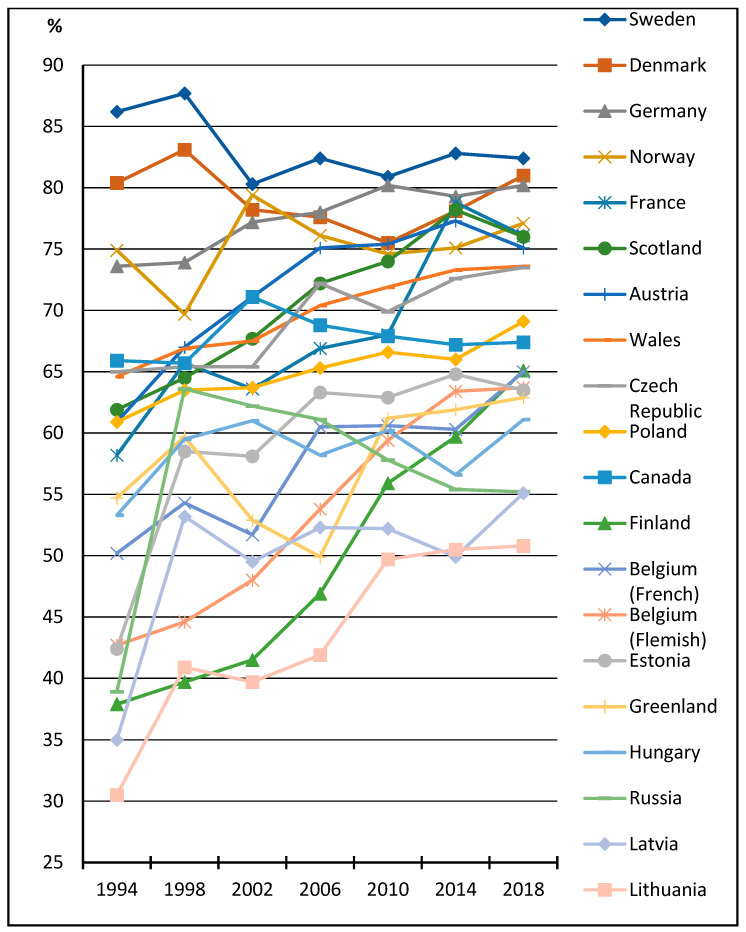
Percentage of students who reported more-than-once-a-day toothbrushing, by survey year and HBSC country.

**Table 1 healthcare-11-03148-t001:** Estimations of linearity in trends in more-than-once-a-day toothbrushing prevalence, by different study periods and HBSC countries.

Country	Estimations								
	1994–2010 *			2010 *–2018			1994–2018		
	OR ^#^	*p*-Value	(99% CI)	OR ^#^	*p*-Value	(99% CI)	OR ^#^	*p*-Value	(99% CI)
Austria	1.20	<0.001	(1.17; 1.23)	0.99	0.569	(0.92; 1.05)	1.13	<0.001	(1.11; 1.15)
Belgium (Flemish)	1.19	<0.001	(1.16; 1.22)	1.10	<0.001	(1.03; 1.16)	1.20	<0.001	(1.18; 1.22)
Belgium (French)	1.13	<0.001	(1.10; 1.16)	1.10	<0.001	(1.04; 1.17)	1.11	<0.001	(1.09; 1.13)
Canada	1.03	<0.001	(1.01; 1.05)	0.99	0.402	(0.96; 1.02)	1.01	0.039	(1.00; 1.02)
Czech Republic	1.08	<0.001	(1.05; 1.11)	1.10	<0.001	(1.05; 1.16)	1.08	<0.001	(1.07; 1.10)
Denmark	0.91	<0.001	(0.88; 0.94)	1.16	<0.001	(1.08; 1.25)	0.97	0.001	(0.95; 0.99)
Estonia	1.24	<0.001	(1.20; 1.27)	1.02	0.355	(0.96; 1.08)	1.14	<0.001	(1.12; 1.16)
Finland	1.21	<0.001	(1.18; 1.24)	1.22	<0.001	(1.16; 1.30)	1.23	<0.001	(1.21; 1.25)
France	1.09	<0.001	(1.07; 1.12)	1.22	<0.001	(1.16; 1.28)	1.17	<0.001	(1.15; 1.19)
Germany	1.10	<0.001	(1.07; 1.14)	1.00	0.945	(0.93; 1.07)	1.07	<0.001	(1.05; 1.09)
Greenland	1.00	0.856	(0.96; 1.05)	1.03	0.524	(0.92; 1.15)	1.05	<0.001	(1.02; 1.08)
Hungary	1.05	<0.001	(1.03; 1.08)	1.02	0.327	(0.97; 1.08)	1.03	<0.001	(1.02; 1.05)
Latvia	1.16	<0.001	(1.13; 1.19)	1.07	0.002	(1.02; 1.13)	1.09	<0.001	(1.08; 1.11)
Lithuania	1.20	<0.001	(1.18; 1.23)	1.02	0.419	(0.96; 1.08)	1.19	<0.001	(1.16; 1.22)
Norway	1.04	0.001	(1.01; 1.07)	1.05	0.062	(0.98; 1.13)	1.03	<0.001	(1.01; 1.05)
Poland	1.06	<0.001	(1.03; 1.09)	1.07	0.006	(1.01; 1.13)	1.06	<0.001	(1.04; 1.07)
Russia	1.15	<0.001	(1.12; 1.18)	0.94	0.003	(0.89; 0.99)	1.03	<0.001	(1.01; 1.04)
Sweden	0.91	<0.001	(0.88; 0.94)	1.07	0.012	(1.00; 1.14)	0.96	<0.001	(0.94; 0.98)
Scotland	1.16	<0.001	(1.13; 1.19)	1.07	0.002	(1.01; 1.14)	1.15	<0.001	(1.13; 1.17)
Wales	1.10	<0.001	(1.07; 1.13)	1.04	0.023	(1.00; 1.09)	1.08	<0.001	(1.07; 1.10)
Overall sample	1.10	<0.001	(1.09; 1.11)	1.06	<0.001	(1.05; 1.07)	1.08	<0.001	(1.07; 1.09)

Notes: * The data of the survey in 2010 were used for both the end of the 1994–2010 period and the beginning of the 2010–2018 period. ^#^ Binary logistic regression model was used to calculate OR for more-than-once-a-day toothbrushing in association with survey wave, which was treated as a continuous variable; gender and age category were adjusting covariates.

**Table 2 healthcare-11-03148-t002:** Trends in more-than-once-a-day toothbrushing prevalence over the period of 1994 to 2018 in the overall HBSC sample and subsamples stratified by gender and age.

Group of Respondents	Prevalence of Tooth Brushing More than Once a Day (99% CI)							Interwave OR ^#^
	1994	1998	2002	2006	2010	2014	2018	
Overall sample	56.9	62.2	62.5	64.7	66.2	67.6	68.7	1.08, *p* < 0.001
	(56.5; 57.3)	(61.8; 62.7)	(62.1; 62.9)	(64.3; 65.0)	(65.9; 66.6)	(67.2; 67.9)	(68.4; 69.1)	(1.07; 1.09) ^a^
Boys:	49.1	53.7	54.6	57.1	59	60.2	61.2	1.08, *p* < 0.001
	(48.5; 49.7)	(53.1; 54.4)	(54.0; 55.1)	(56.5; 57.7)	(58.4; 59.5)	(59.6; 60.8)	(60.7; 61.7)	(1.07; 1.09) ^b^
11 years	48.9	54	55.8	57.2	61.1	62.9	64.8	1.11, *p* < 0.001
	(47.8; 50.0)	(52.9; 55.1)	(54.8; 56.8)	(56.2; 58.2)	(60.1; 62.0)	(62.0; 63.9)	(63.9; 65.7)	(1.10; 1.12)
13 years	48.7	52.8	53.7	56.9	58.5	59	60.1	1.08, *p* < 0.001
	(47.6; 49.8)	(51.7; 53.9)	(52.7; 54.7)	(56.0; 57.9)	(57.5; 59.4)	(58.0; 60.0)	(59.2; 61.0)	(1.07; 1.09)
15 years	49.7	54.5 *	54.2 ***	57.2	57.3 ***	58.7 ***	58.4 ***	1.06, *p* < 0.001
	(48.6; 50.9)	(53.4; 55.6)	53.1; 55.2)	(56.2; 58.2)	(56.3; 58.2)	(57.7; 59.7	(57.4; 59.4)	(1.05; 1.07)
Girls:	64.3	70.4	70	72	73.3	74.7	75.9	1.09, *p* < 0.001
	63.7; 64.8)	(69.8; 71.0)	(69.4; 70.5)	(71.5; 72.5)	(72.8; 73.8)	(74.2; 75.2)	(75.4; 76.3)	(1.08; 1.10) ^b^
11 years	60.4	66.7	66.9	68.8	70.4	72	74.2	1.10, *p* < 0.001
	(59.4; 61.4)	(65.7; 67.7)	(65.9; 67.8)	(67.9; 69.8)	(69.5; 71.2)	(71.1; 72.9)	(73.4; 75.0)	(1.09; 1.11)
13 years	63.7	70.2	69	71.4	72.3	74	76.4	1.09, *p* < 0.001
	(62.6; 64.7)	(69.2; 71.2)	(68.1; 69.9)	(70.5; 72.3)	(71.5; 73.2)	(73.1; 74.8)	(75.6; 77.2)	(1.08; 1.10)
15 years	69.0 ***	74.3 ***	74.4 ***	75.7 ***	77.2 ***	78.2 ***	77.2 ***	1.07, *p* < 0.001
	(68.0; 70.0)	(73.3; 75.2)	(73.6; 75.3)	(74.9; 76.6)	(76.4; 78.0)	(77.3; 79.0)	(76.4; 78.0)	(1.06; 1.08)

Notes: Data were weighted by country sample size. * *p* < 0.05, *** *p* < 0.001 comparing prevalence by age category (Chi-squared test). ^#^ OR for the continuous survey wave, *p*-value, and (99% CI for OR); ^a^ analysis adjusted for gender and age category; ^b^ analysis adjusted for age category.

**Table 3 healthcare-11-03148-t003:** Association of tooth brushing with demographic and social factors, by year of the survey: results from multivariable logistic regression analysis.

Year of the Survey	OR, *p*-Value, and (99% CI)					
	Gender (Girls vs. Boys)	Age Category		Family Affluence Score		Family Structure
		(13 yrs vs. 11 yrs)	(15 yrs vs. 11 yrs)	(Medium vs. Low)	(High vs. Low)	(Non-Intact vs. Intact)
1994	1.85, *p* < 0.001	1.07, *p* < 0.001	1.26, *p* < 0.001	1.25, *p* < 0.001	1.30, *p* < 0.001	0.97, *p* = 0.083
	(1.79; 1.92)	(1.02; 1.12)	(1.20; 1.32)	(1.18; 1.32)	(1.22; 1.38)	(0.93; 1.02)
1998	2.05, *p* < 0.001	1.07, *p* < 0.001	1.22, *p* < 0.001	1.39, *p* < 0.001	2.04, *p* < 0.001	0.91, *p* < 0.001
	(1.97; 2.13)	(1.02; 1.12)	(1.16; 1.28)	(1.32; 1.46)	(1.91; 2.17)	(0.87; 0.95)
2002	1.97, *p* < 0.001	1.01, *p* = 0.462	1.16, *p* < 0.001	1.35, *p* < 0.001	1.79, *p* < 0.001	0.98, *p* = 0.145
	(1.90; 2.04)	(0.97; 1.06)	(1.11; 1.21)	(1.29; 1.42)	(1.69; 1.90)	(0.94; 1.02)
2006	1.96, *p* < 0.001	1.05, *p* = 0.002	1.17, *p* < 0.001	1.39, *p* < 0.001	1.90, *p* < 0.001	0.92, *p* < 0.001
	(1.89; 2.03)	(1.01; 1.10)	(1.12; 1.22)	(1.33; 1.45)	(1.79; 2.01)	(0.88; 0.95)
2010	1.90, *p* < 0.001	1.00, *p* = 0.862	1.09, *p* < 0.001	1.35, *p* < 0.001	1.85, *p* < 0.001	0.90, *p* < 0.001
	(1.84; 1.97)	(0.95; 1.04)	(1.04; 1.13)	(1.29; 1.42)	(1.75; 1.96)	(0.87; 0.94)
2014	1.96, *p* < 0.001	0.97, *p* = 0.076	1.07, *p* < 0.001	1.44, *p* < 0.001	1.97, *p* < 0.001	0.87, *p* < 0.001
	(1.89; 2.03)	(0.93; 1.01)	(1.03; 1.12)	(1.38; 1.51)	(1.86; 2.10)	(0.84; 0.91)
2018	2.03, *p* < 0.001	0.94, *p* < 0.001	0.94, *p* < 0.001	1.44, *p* < 0.001	2.01, *p* < 0.001	0.85, *p* < 0.001
	(1.95; 2.10)	(0.90; 0.98)	(0.89; 0.98)	(1.37; 1.50)	(1.89; 2.13)	(0.81; 0.88)

Notes: Data were weighted by country sample size.

## Data Availability

The data presented in this study are available upon reasonable request from the HBSC Data Management Centre, University of Bergen, Norway (dmc@hbsc.org).

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
