# Peer review of "Adolescent Toothbrushing and Its Association with Sociodemographic Factors—Time Trends from 1994 to 2018 in Twenty Countries"

_healthcare, 2023, doi:10.3390/healthcare11243148_

Round 1
Reviewer 1 Report
Comments and Suggestions for Authors
Dear Authors,
I have been invited to review your work entitled “Adolescent Toothbrushing and its Association with Sociodemographic Factors – Time Trends from 1994 to 2018 in Twenty Countries”. I believe it is a work of concern, however, there are minor issues that deserve revision for the acceptance of this work to the Healthcare (MDPI).
Please, provide a point-by-point response, highlighting the corrections with a color mark specific to each reviewer.
Introduction
The introduction is properly written, including an initial brief description of the topic bringing the focus to the topic of interest. Finally, the objectives have been correctly described.
However, the last sentence "In addition, we anticipate some changes in association between toothbrushing and sociodemographic factors may have occurred over time." should be removed since it anticipates a possible outcome that is described later.
Finally, where you have written "the aim of this study," it would be more helpful to readers if you specified the type of study (e.g., retrospective observational study).
Materials and Methods
This section contains all the necessary information, indicating the population of interest, the correct explanation regarding the non-need for the ethics committee, the variables of interest, and the statistical analysis.
In addition, could you please include the HBSC survey in the supplementary materials? It might be helpful for readers to have it easily available when reading your manuscript.
Results
The part about results is well described, pointing to a figure that graphically represents the percentage of students who reported more-than-once-a-day toothbrushing and tables inherent in the inferential statistics performed. Their description in the text is correctly executed. I have nothing more to add for this section.
Discussion and Conclusions
Appropriate and comprehensive comparisons were made between the results of this manuscript and other studies in the literature. In addition, a description of what was revealed by the statistical analysis was well executed. The limitations of the present study and future studies needed to increase knowledge of this topic are present. No other additions are present in my opinion to be performed.

Reviewer 2 Report
Comments and Suggestions for Authors
Although the relationship between regular tooth brushing and socioeconomic factors has been reported in several studies, this study is significant in that it analyzed long-term temporal trends using youth data from 20 countries. The manuscript is structured well and systematically. However, some minor details need to be checked.
In the introduction section, the research objectives according to the research needs are clear.
The methods section presents the research subjects, research variables, and analysis methods in detail.
In the results section, the year 2010 overlaps in the year range classification in Table 1: ‘1994-2010’ and ‘2010-2018’. Shouldn’t ‘2010-2018’ become ‘2011-2018’? Please check.
Although the clinical significance and limitations of this study are specifically presented in the discussion section, please add future research directions.

Reviewer 3 Report
Comments and Suggestions for Authors
This study presents an intriguing exploration of trends in frequent toothbrushing (more than once a day) among adolescents, drawing on data from HBSC surveys conducted from 1994 to 2018. While the study contributes valuable insights to the field, there are a few areas where minor revisions could significantly enhance the manuscript's readability and overall quality.
Abstract:
The abstract should detail the number of participants in the study to provide a comprehensive overview of the research scale and context.
Materials and Methods:
To better inform the reader, it is recommended to include a figure or textual description listing the 20 countries included in the study. This addition will improve understanding and provide a clear reference to the study's geographical scope.
Line 116 Revision:
It is suggested that the term “toothbrushing frequency as an output variable” be revised to “toothbrushing frequency as an outcome variable,” aligning with more standard research terminology.
Line 138 Clarification:
The manuscript should describe in greater detail the primary outcome variable of the study, including how it was collected, the specific question posed, and whether this question remained consistent across all surveyed countries and waves.
Results:
For Table 3, please delineate the reference group for each variable to provide readers with a clear understanding of the baseline comparisons being made.
Discussion:
The discussion should commence with a clear statement of the study's main objective, followed by a concise summary of the principal findings to immediately convey the research's purpose and conclusions to the reader.
Reviewer 4 Report
Comments and Suggestions for Authors
The paper represents a useful research analysis on OH habits in some crucial categories as teenagers and the novelty in the paper is linking the TB habit with FAS and the family structure. I would spend a bit more describing the statistical methods which have been used and a bit more about the impact in terms of prevention this analysis would cause. Other than that I would work a bit around the presentation, maybe increasing the number of graphs.
Comments on the Quality of English LanguageA moderate English check is needed
Reviewer 5 Report
Comments and Suggestions for Authors
This study is well-conducted and designed. However, in writing some points are confusing and must be explained and addressed by the authors.
- Indicate the study’s design with a commonly used term in the title or the abstract.
- The introduction is well done.
- Please, present key elements of study design early in the paper.
- Clearly define all outcomes, exposures, predictors, potential confounders, and effect modifiers.
- For each variable of interest, give sources of data and details of methods of assessment (measurement).
- Explain how the study size was arrived at
- Explain how quantitative variables were handled in the analyses.
- Describe all statistical methods, including those used to control for confounding
- Describe analytical methods taking account of the sampling strategy
- Discuss the generalisability (external validity) of the study results
- In the discussion section, please, summarise key results with reference to study objectives.
- Discuss limitations of the study, taking into account sources of potential bias or imprecision.
- Discuss both the direction and magnitude of any potential bias.
